# Development of a Tetraplex qPCR for the Molecular Identification and Quantification of Human Enteric Viruses, NoV and HAV, in Fish Samples

**DOI:** 10.3390/microorganisms9061149

**Published:** 2021-05-27

**Authors:** Andreia Filipa-Silva, Mónica Nunes, Ricardo Parreira, Maria Teresa Barreto Crespo

**Affiliations:** 1ITQB NOVA, Instituto de Tecnologia Química e Biológica António Xavier, Universidade Nova de Lisboa, Av. da República, 2780-157 Oeiras, Portugal; andreiasilva@ibet.pt (A.F.-S.); tcrespo@ibet.pt (M.T.B.C.); 2iBET, Instituto de Biologia Experimental e Tecnológica, Apartado 12, 2780-157 Oeiras, Portugal; 3Global Health and Tropical Medicine (GHTM) Research Center, Unidade de Microbiologia Médica, Instituto de Higiene e Medicina Tropical (IHTM), Universidade Nova de Lisboa (NOVA), 1349-008 Lisboa, Portugal; ricardo@ihmt.unl.pt

**Keywords:** pathogenic human viruses, fish, tetraplex qPCR assay, norovirus, hepatitis A

## Abstract

Human enteric viruses such as norovirus (NoV) and hepatitis A virus (HAV) are some of the most important causes of foodborne infections worldwide. Usually, infection via fish consumption is not a concern regarding these viruses, since fish are mainly consumed cooked. However, in the last years, raw fish consumption has become increasingly common, especially involving the use of seabass and gilthead seabream in dishes like sushi, sashimi, poke, and carpaccio. Therefore, the risk for viral infection via the consumption of raw fish has also increased. In this study, a virologic screening was performed in 323 fish specimens captured along the Portuguese coast using a tetraplex qPCR optimised for two templates (plasmid and in vitro transcribed RNA) to detect and quantify NoV GI, NoV GII and HAV genomes. A difference of approximately 1-log was found between the use of plasmid or in vitro transcribed RNA for molecular-based quantifications, showing an underestimation of genome copy-number equivalents using plasmid standard-based curves. Additionally, the presence of NoV genomic RNA in a pool of seabass brains was identified, which was shown to cluster with a major group of human norovirus sequences from genogroup I (GI.1) by phylogenetic analysis. None of the analysed fish revealed the presence of NoV GII or HAV. This result corroborates the hypothesis that enteric viruses circulate in seawater or that fish were contaminated during their transportation/handling, representing a potential risk to humans through raw or undercooked fish consumption.

## 1. Introduction

Enteric viruses represent a major risk to human health, being responsible for numerous outbreaks worldwide. The best characterised foodborne viral agents are human norovirus (NoV) and hepatitis A virus (HAV), which cause the most significant part of foodborne-associated illness globally. According to Centers for Disease Control and Prevention (CDC) and the European Centre for Disease Prevention and Control (ECDC), NoV-associated foodborne infections are one of the most frequently reported in the United States and European Union (EU), being, for instance, associated with 457 outbreaks, and, most importantly with 11,125 cases of illness in 2019 (22.5% of total cases) only in the EU [1]. Moreover, NoV and HAV have been estimated to impart high economic losses, mainly associated with the measures taken to reduce their impact on population health [2,3]. It is expected that foodborne infections cost between USD 55 and USD 93 billion per year in the United States [4], while studies in the Netherlands reported economic losses associated with NoV and HAV to be around EUR 90 million and EUR 2.9 million, respectively [5].

Domestic and restaurants settings are described as the most common places associated with NoV/HAV outbreaks [1], and the majority of cases are attributed to food handling and poor hygiene practices [3]. While, some food matrices, such as fresh vegetables, fruit, and shellfish, which are consumed mostly raw or undercooked, are more susceptible to virus contamination than others, being more frequently associated with foodborne outbreaks, any type of food, especially if consumed raw or undercooked, could be implicated in one such outbreak since contaminated food items can be traded globally and used in a variety of dishes [3].

Usually, since fish are mainly consumed cooked, their ingestion is not a concern regarding a possible contamination with the most common foodborne-associated viruses. However, in recent times raw fish consumption has become increasingly common, and the use of gilthead seabream, Atlantic horse mackerel, and seabass in dishes like sushi, sashimi, poke, and carpaccio has increased. Tthese eating habits create concerns regarding the safety of consuming raw fish since some may be contaminated with several types of potentially pathogenic microorganisms, including bacteria, viruses, and parasites [6,7,8]. Altogether, pathogenic microorganisms are responsible for outbreaks of human disease that can either have an environmental origin or result from cross-contamination of food items during their handling [9]. In this context, it is of utmost importance to develop reliable, rapid, and robust methodologies to detect pathogenic human viruses, such as NoV and HAV in food matrices. Most methods currently used for virus detection in food are based on molecular methods involving partial virus genome amplification by real-time (or quantitative) PCR (qPCR). Although virus control is not mandatory, in Europe, an ISO technical standard specification is available to quantify NoV and HAV genomes in foodstuffs (soft fruit, leaf, stem, and bulb vegetables and bottled water) or on food surfaces [10]. However, this ISO method is not validated for viral quantification in fish since different food matrices may interfere with the viral elution efficiency, and organic and inorganic substances that can be present may interfere with the target sequences detection by qPCR. Additionally, this ISO recommends the analysis of one target virus at a time, which can be time consuming and work intensive.

The existing literature suggests that in the assessment of a qPCR assay several factors should be considered, namely, the effect of using different templates to generate standard curves for the absolute quantification of RNA viruses, which can be made from either a plasmid, a synthetic oligonucleotide, or an in vitro transcribed RNA [11]. The advantage of using in vitro transcribed RNA as a template is that it implicates cDNA synthesis, and the efficiency of the reverse transcription reaction can be considered [11,12]. On the other hand, it involves in vitro RNA and cDNA synthesis steps, which are time consuming and expensive [11,12]. Alternately, plasmids which are relatively cheap and easy to generate can be used as a template, however, it does not account for the cDNA synthesis step, a requirement for the amplification of a viral RNA transcript [11].

Therefore, in this study, we developed a tetraplex qPCR assay to detect and quantify NoV GI, NoV GII, and HAV genomes, using mengovirus as an internal control. This method was implemented either using plasmid or an in vitro transcribed RNA for standard curve construction, which allowed for NoV and HAV genomic quantification to be evaluated by comparing the two standards curve types in terms of efficiency, sensitivity, and detection limit. These tetraplex qPCR assays were used to perform a virological screening in fish captured and farmed along the Portuguese coast, focusing on four of the most economically important species from the Atlantic coast. These included gilthead seabream (*Sparus aurata*) and seabass (*Dicentrarchus labrax*), two of the most extensively wild-caught and farmed species in aquaculture, as well as sardine (*Sardina pilchardus*) and Atlantic horse mackerel (*Trachurus trachurus*), two of the most consumed fish species in Portugal.

## 2. Materials and Methods

### 2.1. Process Control Virus

A nonvirulent mutant strain of mengovirus (vMC0) was used as process control since this virus is not naturally present in food matrices [2]. Moreover, mengovirus is a member of the *Picornaviridae* family, sharing structural similarities with HAV. Therefore, it is normally used as a process control virus to detect HAV and NoV in food [13]. Mengovirus was replicated in HeLa cells (ATTCC, CCL-2) by the Analytical Services Unit of iBET (iBET, Oeiras, Portugal) as described by Costafreda et al. (2006) [13]. Total RNA was extracted with the QIAamp Viral RNA Mini Kit (Qiagen, Hilden, Germany) and quantified by measuring the absorbance at 260/280 nm with a NanoDrop ND-1000 spectrophotometer. According to the manufacturer’s instructions, RNA was converted to cDNA in a final volume of 20 μL with the NZY First-Strand cDNA Synthesis Kit (NZYTech, Lisbon, Portugal). This kit included a combination of random hexamers and oligo(dT)18 primers to increase the sensitivity of the reverse transcription reaction. Partial vMC0 genomic amplification was performed by qPCR (see Section 2.5). Based on this approach, the production stock of vMC0 had titres of approximately 10^9^ genome copies/µL.

To determine the optimal input for the control process virus, different amounts of mengovirus were used to spike one soft and internal tissue (liver) and one hard and external tissue (gills) to assess the limit of viral genome detection.

Additionally, before total RNA extraction, all the samples were spiked with 25 µL of mengovirus suspension to indirectly estimate losses of the target viruses, which can occur at several stages during processing. The recovery rate of mengovirus was calculated using the following formula: % recovery control virus = (amount of control virus after extraction/initial amount of control virus in the samples) × 100.

### 2.2. Sample Processing and Total RNA Extraction

A total of 323 fishes were analysed (post-mortem) in this study. These were either (i) wild specimens caught along the coast of Peniche, Figueira da Foz and Algarve, (ii) available at supermarkets, (iii) discarded from fish markets, or (iv) farmed in the Algarve and Setúbal region (Figure 1, Table 1). All the analysed specimens were divided into a total of 50 pools according to their source, species, and tissue type. Each pool was organised according to fish size, including five specimens of the bigger (gilthead seabream and seabass), and 10 specimens of the smaller (sardine and Atlantic horse mackerel) fish, respectively. The fish acquired from supermarkets and discarded from fish markets were combined only in one pool each due to the fewer specimen numbers. From each fish, eight tissue samples were selected, taking into consideration the tissues frequently involved in viral contamination/infection and included (i) eyes, (ii) brain, (iii) gills, (iv) skin, (v) muscle, (vi) liver, (vii) spleen, and (viii) kidney. In total 400 pools (50 pools × 8 organs) were created.

Approximately 2.0 g of fish tissue was chopped using a sterile razor blade and homogenised in 10 mL of TNE buffer (50 mM Tris-HCl, 100 mM NaCl, 0.1 mM EDTA, pH 7.6) using a Precellys Evolution Homogenizer (Bertin Instruments, Montigny-le-Bretonneux, France). The homogenates were centrifuged 10 min at 2000 rpm at 4 °C to remove particulate debris, and the supernatant used for total RNA extraction, carried out from 250 µL of clarified supernatant, using NZYol (NZYTech, Lisbon, Portugal), as described by the supplier. RNA was dissolved in 30 µL of DEPC-water, and the concentration and purity of the obtained RNA extracts determined using a NanoDrop 1000 spectrophotometer. RNA extracts were stored at −80 °C until further use. For qPCR reactions, the RNA was converted to cDNA as described in Section 2.1.

### 2.3. Taqman Probes and Primers

For each target, a distinct set of primers/probes (Table 2) was used. Some sets had been previously published in the literature, while others were designed during this study. For that, reference sequences for each viral targeted gene were retrieved from GenBank, and multiple sequence alignments were created using Mafft 7 [14]. Primers and probes were designed using Multiple Primer Analyser (Thermo Fisher Scientific, Waltham, MA, USA) and PrimerBlast [15].

### 2.4. qPCR Standard Curve Construction

For the construction of the standard curves based on plasmid DNA reference templates, three recombinant plasmids were artificially synthesised (NZYtech, Lisbon, Portugal) harbouring partial sequences of the junction of open reading frames 1 and 2 (ORF1/ORF2) of NoV GI and GII, and the 5′ noncoding region (5′NCR) of HAV. These same plasmids served as a starting point to obtain PCR products from which the in vitro transcribed RNA of each targeted genomic region was subsequently synthesised. Amplicons corresponding to sections of recombinant plasmid DNA were amplified by conventional PCR, and the success of the amplification process was confirmed by their visualisation on 2% agarose gels, followed by their purification with NZYGelpure kit (NZYTech, Lisbon, Portugal). The primers used to obtain these PCR fragments were chosen to include a T7 RNA polymerase-specific promoter upstream of the viral coding sequence. The fragments were then used as a template for in vitro transcription with the NZY T7 High Yield RNA Synthesis kit (NZYTech, Lisbon, Portugal). The obtained RNA was purified with NZY RNA isolation kit (NZYtech, Lisbon, Portugal) and treated with DNase I solution to prevent DNA contamination. RNA concentrations and purity were estimated using a NanoDrop 1000 spectrophotometer. Total RNA was converted to cDNA as previously described and used to construct the quantification standard curves.

Standard curves were constructed (one for each template) using 10-fold serial dilutions ranging from 10^7^ to 10 genome equivalents. Each target was either tested individually or as a mixture.

### 2.5. Single and Multiplex qPCR Assay

Single and multiplex qPCR reactions for detecting NoV GI, NoV GII, HAV, and mengovirus were carried out in a total volume of 20 μL using SensiFAST™ Probe No-ROX amplification mix (Bioline, Memphis, TN, USA). The concentrations of Nov GI, NoV GII, HAV, and mengovirus forward/reverse primers and probes (Table 3), as well as qPCR temperature profile for single and multiplex qPCR assays (5 min at 95 °C as hot-start, and 40 cycles of 15 s at 95 °C for denaturation, 1 min at 60 °C for annealing, and 1 min at 65 °C for extension), were established based on the study of Fuentes et al. (2014) [2]. Negative controls containing nuclease-free water were included in each run to rule out the possibility of false-positive amplification results due to cross-contamination. Thermal cycling, fluorescent data collection, and data analyses were performed in a LightCycler 96 real-time PCR System (Roche, Basel, Switzerland), according to the manufacturer’s instructions.

### 2.6. Analytical Specificity and Detection Limit Evaluation for the Single and Multiplex qPCR Assays

All single and multiplex assays for the detection of NoV GI, NoV GII, HAV, and mengovirus genomes were tested for cross-reactivity with other viruses available including, adenovirus type 5 (HAdV-5; family *Adenoviridae*), infectious pancreatic necrosis virus (IPNV; family *Birnaviridae*), infectious hematopoietic necrosis virus (IHNV; family *Rhabdoviridae*), viral haemorrhagic septicaemia virus (VHSV; family *Rhabdoviridae*), viral nervous necrosis virus (VNNV; family *Nodaviridae*) and Hepatitis E virus (HEV; family *Hepeviridae*).

For evaluating the sensitivity of each assay, viral sequences were detected in serial dilutions of plasmid or cDNA prepared as mentioned in Section 2.4. The qPCR assay amplifications were carried out either using each plasmid/cDNA dilution or a mixture of all.

#### Evaluation of the Analytical Specificity and Detection Limit of the qPCR Assays with Previously Positive Samples

Using the two types of standard curves (plasmid and in vitro transcribed RNA) for viral quantification, positive wastewater samples previously shown to contain NoV GI, NoV GII, and HAV genomes (our work—not published), were used to validate the qPCR assays developed. The quantification of genome equivalents was carried out by qPCR using either plasmid or in vitro transcribed RNA standard-based curves. RNA was extracted from these samples with QIAamp Viral RNA Mini Kit (Qiagen, Hilden, Germany) according to the manufacturer’s instructions, using 140 µL of each sample and an elution volume of 80 µL (double elution 2 × 40 µL). RNA concentration and purity were estimated using a NanoDrop 1000 spectrophotometer. RNA was converted to cDNA as previously described in Section 2.1.

### 2.7. Nested PCR Assays for NoV GI, NoV GII, and HAV Detection

Three nested-PCR (nPCR) protocols targeting NoV GI, NoV GII, and HAV genomes were developed to further characterise by phylogenetic analysis the positive samples previously obtained by the qPCR protocols. All nPCR assays were optimised using synthetic templates purchase from American Type Culture Collection (ATCC, Manassas, Virginia, USA), including Quantitative Synthetic Norovirus G1 (I) RNA ATCC^®^ VR-3234SD™, Quantitative Synthetic Norovirus G2 (II) RNA ATCC^®^ VR3235SD™, and Quantitative Synthetic Hepatitis A virus DNA VR-3257SD™. NoV GI and HAV detection was carried out in a total volume of 25 μL using NZYTaq II 2× Green Master mix (Nzytech, Lisbon, Portugal). For NoV GI, each 25 µL reaction volume included 0.4 µM of each primer (Table 2) and 5 µL of cDNA template. The cycling conditions were 94 °C for 3 min, followed by 35 cycles of amplification with an initial denaturation at 94 °C for 1 min, primer annealing at 45 °C for 1 min, and primer extension at 72 °C for 1 min, followed by a final 10 min extension step a 72 °C. The product of the first reaction was used as a template in the second round of the nPCR reaction, which was performed at 94 °C for 3 min, followed by 30 cycles of amplification with an initial denaturation at 94 °C for 1 min, primer annealing at 45 °C for 1 min, and primer extension at 72 °C for 1 min. These steps were followed by a final 10 min extension step at 72 °C. For HAV detection, each 25 μL of reaction included 10 µM of each primer (Table 2) and 5 µL of cDNA template using previously described cycling conditions [20].

For NoV GII detection, the NZYTaq II 2× Green Master mix was not efficient, being used the Platinum™ SuperFi II Green PCR Master Mix (Invitrogen ™, Thermo Fisher Scientific, Waltham, MA, USA) in a total volume of 20 μL. The thermal cycling conditions for PCR and nPCR included 1 cycle at 98 °C for 30 s, followed by 35 cycles at 98 °C for 10 s, 60 °C for 10 s and 72 °C for 30 s, and a 10 min final extension step at 72 °C.

All amplification steps were performed on a Doppio VWR thermocycler (VWR, Monroeville, PA, USA), and their success was confirmed by amplicon visualisation on 1.5% agarose gels. Additionally, non-template (negative) controls were used in each run. PCR products were sequenced in both directions using Sanger’s method (Eurofins Genomics, Ebersberg, Germany with nPCR primers. The search of homologs in the public genetic databases was carried out with the NCBI Basic Local Alignment Search Tool (https://blast.ncbi.nlm.nih.gov/Blast.cgi (accessed on 31 March 2021)).

### 2.8. Dataset Compilation and Phylogenetic Analysis

Nucleotide (nt) sequences used for the preparation of the different sequence datasets were selected among those deposited in the GenBank database, on the proviso that they would be representative of (i) each of the previously described species with (ii) a significant sequence overlap with the sequences obtained during this study to maximise the number of unambiguously aligned nt positions in each sequence alignment. For phylogenetic analysis, multiple alignments of nt sequences were constructed with the iterative G-INS-I method as implemented in MAFFT vs. 7 [14], followed by their edition using GBlocks [21]. Phylogenetic trees were constructed using the maximum likelihood (ML) optimisation criterium and the best fitting evolutionary model (GTR+Γ+I; GTR—General Time Reversal, Γ—Gamma distribution, I—proportion of invariant sites), as suggested W-IQ-TREE [22]. Phylogenetic reconstructions were carried out using IQ-TREE version 2.1.2 for MacOSX [22], and the stability of the obtained ML tree topologies assessed by bootstrapping with 1000 re-samplings of the original sequence data.

## 3. Results

### 3.1. Multiplex qPCR Implementation

#### 3.1.1. qPCR Efficiency, Analytical Specificity, and Sensitivity

The generation of a standard curve for qPCR quantification of RNA viruses can be constructed based on the use of serial dilutions of several possible templates, including a viral genome, a structurally equivalent template prepared from in vitro transcribed RNA, or a plasmid harbouring the targeted sequence in the form of dsDNA. In an ideal situation, standard curves for qPCR analysis should be constructed using serial dilutions of cDNA prepared from RNA extracted from a viral suspension. However, since logistic limitations deter virus isolation/propagation in our laboratory, plasmids and in vitro transcribed RNA molecules prepared from a synthetic template were chosen as the starting model for standard-curve construction.

The standard curve for single and multiplex qPCR reactions for detecting NoV GI, NoV GII, HAV, and mengovirus genomes was validated by the parameters listed in Table 4 and Appendix A. Overall, both qPCR reactions (single and multiplex) presented a good determination coefficient (squared), ranging between 0.985 and 1. Moreover, their efficiencies, calculated from the slope of the obtained standard curves (see the legend to Figure 2), varied between 88% and 111%. Detection limits for NoV GII and HAV were 10 genome copies, calculated when a plasmid template was used for the standard curve construction in the single and multiplex qPCR assay. On the other hand, the NoV GI genome detection limit in the multiplex format was approximately 1-log higher (100 copies per reaction) than in the single format (Figure 2). Regarding the standard curves obtained with the in vitro transcribed RNA, the detection limits for NoV GI and NoV GII genomes were similar in single and multiplex assays, with sensitivities of 10^3^ and 10 genome copies, respectively. For HAV, the detection limit was 10^2^ genome copies in the single reaction, increasing 1-log in the multiplex format (Figure 2).

Overall, the singleplex/multiplex qPCR assays described presented specificity, as unspecific amplifications were not detected with the templates described in Section 2.6.

#### 3.1.2. Quantification of the Wastewater Samples Positive for NoV GI, NoV GII, and HAV Genomes Using a Plasmid and an In Vitro Transcribed RNA Standard-Based Curve

The wastewater samples positive for NoV GI, NoV GII, and HAV genomes used to validate the qPCR assays developed during this study gave similar Cq (cycle quantification) values between samples using the plasmid and the in vitro transcribed RNA standard-based curve (Table 5). In terms of sample quantification, a difference of approximately 1-log was found between the plasmid and the in vitro transcribed RNA-based quantification, showing an underestimation using the plasmid standard-based curve (Figure 2 and Figure 3).

### 3.2. Quantification of Mengovirus in Artificially Spiked Samples

Mengovirus detection and quantification in samples spiked with 25 µL of a mengovirus suspension initially titered at 10^9^ genome copies/µL, gave similar multiplex qPCR quantification results (Table 6) regardless of the method used for the construction of a calibration curve (plasmid or cDNA prepared from in vitro transcribed RNA).

### 3.3. Quantification and Characterisation of Human Pathogenic Viruses in Fish

The analysis of the 323 fish specimens using plasmid as a standard curve for NoV and HAV detection and quantification, revealed one pool where the presence of 1.55 × 10^3^ genome copies (Cq = 32.56) of NoV GI genome was disclosed, corresponding to a pool of seabass brain tissue from specimens obtained from Peniche fish market (on the Central Atlantic coast). Similarly, using in vitro transcribed RNA as a standard curve, this sample was also positive with a Cq value of 32.89 corresponding to 4.64 × 10^4^ genome copies of NoV GI. None of the 323 fish specimens analysed revealed the presence of NoV GII or HAV genomes.

The positive sample for NoV GI obtained by qPCR was further confirmed by conventional nPCR and characterised by Sanger sequencing of the open reading frame 1 (ORF1)-ORF2 junction region, the most conserved region of the norovirus genome [18]. The norovirus sequence detected in this study showed between 91 and 98% sequence similarity with the sequences described by the accession numbers KF039737 and KT732279 isolated in the USA and China, correspondingly. Moreover, the phylogenetic analysis revealed that the norovirus nucleotide sequence detected in the brain of seabass was allocated in genogroup I, genotype 1 (accession number LC627095) (Figure 4).

## 4. Discussion

NoV and HAV are the leading cause of foodborne viral infections, being responsible for considerable economic and health burdens globally. Moreover, of all the viruses associated with foodborne infections, NoV and HAV are the most important viral pathogens regarding the severity of the associated illnesses and their common occurrence worldwide [23]. They are transmitted as a result of consumption of not only contaminated food and water but also through direct contact with infected individuals and environmental surfaces exposed to these viruses. In this context, reliable and affordable methodologies for the detection of these viruses are of utmost importance since one of the most efficient ways to prevent and control foodborne infections relies on the implementation of surveillance systems that use rapid, sensitive, and robust diagnostic methods, allowing the prompt identification of pathogens.

In the present study, a virologic screening of the two above-mentioned enteric viruses targeted mainly four fish species highly consumed in Europe and caught/farmed along the Portuguese coast. For that purpose, we used a multiplex qPCR assay designed for the detection/quantification of NoV and HAV genomes, considering two different approaches as standard curves: an external in vitro synthesised RNA and a plasmid. Furthermore, this assay also included a control virus since losses of the targeted viral genomes can occur at several stages during sample preparation and nucleic acid extraction [24,25]. All samples were spiked with a defined amount of a reference virus (mengovirus vMC0) prior to nucleic acid extraction to account for these losses. The control virus used is not expected to occur naturally in the foodstuffs under test, and the efficiency of extraction should be superior to 1% [24,25]. Therefore, we used as control a nonvirulent strain of vMC0, titered at 10^9^ genome copies/µL, and in all the performed extractions, its recovery rate was consistently above 10%.

Multiplex qPCR-based methods have been designed to detect and quantify several targets in a single reaction. However, they tend to display lower detection sensitivity when compared to standard singleplex qPCR [2]. These losses in sensitivity tend to correlate with the number of targets included in a qPCR reaction and the number of copies of each target [2]. However, despite this decrease of sensitivity, which in the present study was within the range of other described multiplex assays [2,26], multiplex qPCR assay does provide considerable savings in cost, reagents, and time. The critical balance between loss of sensitivity and cost/time reductions should be considered [2], since the use of tetraplex protocols can be useful as a routine protocol not only for food monitoring, but also for viral screenings as the one performed in this study.

The standard curve in an absolute qPCR assay is generated by amplifying serial dilutions of a standard DNA, which can be a plasmid, a PCR amplicon, a synthesised oligonucleotide, a genomic DNA, or a cDNA. Among the various types of the standard template, using a plasmid is one of the most common options due to its high stability and preparation reproducibility. However, plasmids could adopt several conformational structures, namely a supercoiled form, which can suppress qPCR assay compared to other templates [27,28]. In this study, we observed that the multiplex qPCR with the in vitro transcribed RNA-based standard curve presented a higher sensitivity in approximately one logarithmic unit than the qPCR using the plasmid-based standard curve. This difference could be due to the lower efficiency of amplification using circular plasmid as a template, especially in the early stage of PCR when it is the dominant template, as previously mentioned [27]. On the other hand, in vitro transcribed RNA-based standard curves may present some limitations since RNA stability could be a source of variability in the final analysis. In fact, in vitro transcription of the amplicon of interest and the reverse-transcription has a great impact in the absolute quantification, being affected by many factors (e.g., enzymes, inhibitors such salts or phenol, and temperature of primer hybridisation and cDNA synthesis, or even the possible formation of secondary structures in RNA molecules), that may impact the results of a qPCR assay [12,29]. Additionally, the preparation of artificial RNA standards could be a work-intensive process since it involves the construction of plasmids with the amplicon of interest that must be in vitro transcribed into RNA, accurately quantified, and converted to cDNA via reverse transcription. Despite all this, RNA standards may help generate more accurate copy number data since they are a better approximation of the RNA viruses present in biological samples [12,27].

Interestingly, in this study, NoV GI genomic RNA was detected in one out of a total of 400 pools (50 pools of fish × 8 organs). Although seafood contamination by these viruses is well described [30,31,32,33,34,35], this result was unexpected, since to our knowledge, this is the first report of norovirus genome in the brain of a seabass. There have been a few reports of acute encephalitis/encephalopathy in humans associated with NoV infection [36,37], which could explain the presence of this virus in this tissue. Furthermore, NoV replication has been reported in several animals such as chimpanzees, gnotobiotic pigs, calves [38], and more recently in zebrafish, where NoV GI and GII replication was observed without visible signs of disease [39]. Although we could not discard contamination during sample processing and preparation, this seems unlikely since no other tissue from these fish was positive for the NoV GI genome.

As expected, the phylogenetic analysis of the norovirus sequence detected in this study revealed that it clustered with a major group containing norovirus sequences from genogroup I, in particular GI.1 (Figure 4), since it is well known that the most common norovirus genogroups detected in patients worldwide are genogroups, GI and GII, being each of them further subdivided into genotypes (9 GI, 27 GII) [40].

The detection of NoV genomes in fish samples can indicate that the water where the analysed fish swam was contaminated with human faeces, probably due to sewage pollution. Despite the fact that the most common categories linked to outbreaks of NoV and HAV are fresh vegetables, fruit, and shellfish since they are consumed raw or undercooked, any type of food could be implicated in an outbreak [3]. Information about outbreaks associated with these enteric viruses in fish is scarce since identifying the food vehicles in an outbreak is not always possible [41]. Nonetheless, in 2019, fish and fishery products were implicated in 193 outbreaks in the EU, being 145 of those caused by noroviruses and other caliciviruses [1]. Additionally, CDC compiles searchable lists of norovirus outbreaks that can be retrieved from the National Outbreak Reporting System (NORS; https://wwwn.cdc.gov/norsdashboard/ (accessed on 2 April 2021)). In this system regarding the years between 2007 and 2018, norovirus outbreak data showed that fish as a vehicle of transmission of NoV infection, without accounting for person-to-person transmission, were responsible for 15 outbreaks with 180 infected people and four hospitalisations. Moreover, fish contamination could be due to the presence of enteric viruses in the surrounding water. In fact, waterborne infections due to noroviruses were responsible for 55 outbreaks, 3330 illnesses and 33 hospitalisations (NORS; https://wwwn.cdc.gov/norsdashboard/ (accessed on 16 May 2021)) between 2007 and 2018 in the United States, whereas in the EU, in 2019, were responsible for 11 outbreaks and 26 hospitalisations [1].

In conclusion, we developed a multiplex qPCR method to identify and quantify NoV and HAV genomes based on two different standard-curve construction approaches. Our data suggest that the one based on in vitro transcribed RNA is a better solution for RNA virus quantification in biological samples. Furthermore, detectingthe human pathogenic NoV GI genome in fish brains supports the hypothesis that these viruses circulate in seawater. On the other hand, while foodborne infection control might be very straightforward by simply cooking the food before consumption, raw or undercooked fish consumption as a common trend makes this control very difficult. Thereby, the monitorisation of these pathogens to properly assess human health risks associated with raw or undercooked fish consumption could be of utmost importance since NoV and HAV are characterised by a high rate of transmission, which makes them even more challenging to control.

## Figures and Tables

**Figure 1 microorganisms-09-01149-f001:**
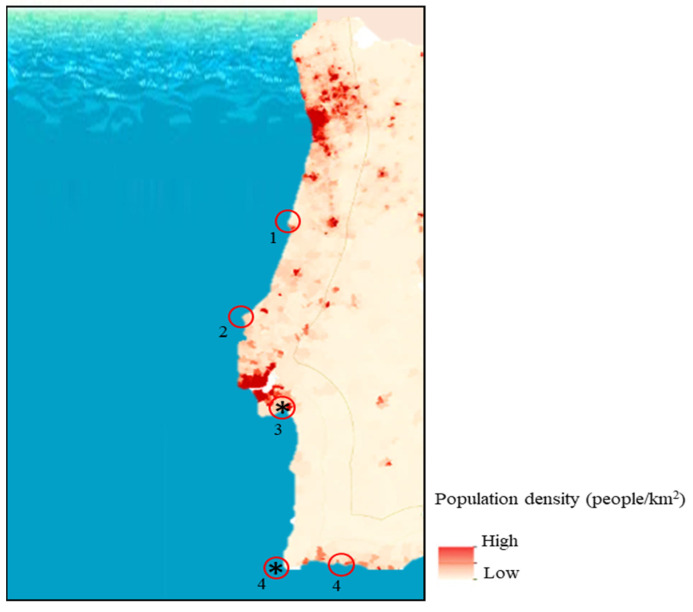
Population density on the Portuguese coast and localisation of the studied fish sampling regions (adapted from https://www.portugal.gov.pt/pt/gc21/governo/programa/programa-nacional-para-a-coesao-territorial-/ficheiros-coesaoterritorial/programa-nacional-para-a-coesao-territorial-o-interior-em-numeros-territorio-pdf.aspx (accessed on 13 February 2021)); * indicates aquaculture sampling sites.

**Figure 2 microorganisms-09-01149-f002:**
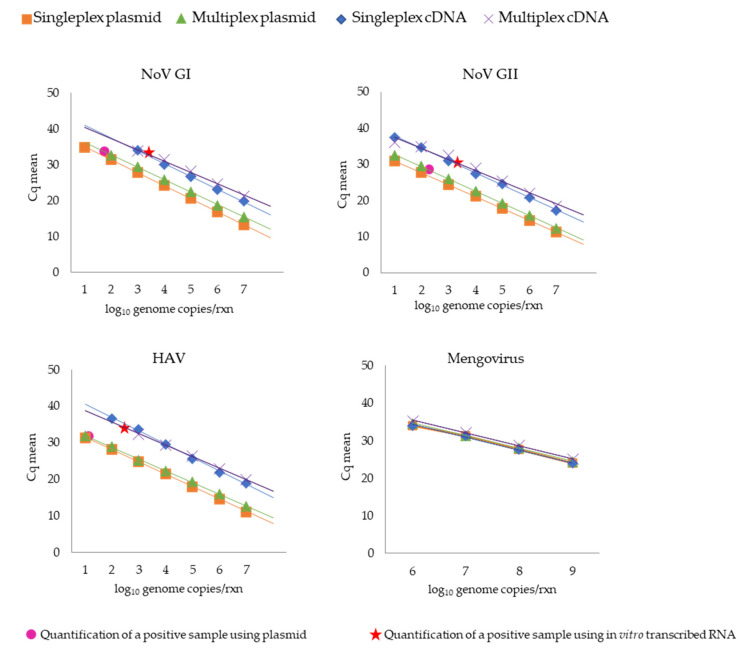
Comparative standard curves of single and multiplex qPCR assays targeting plasmid and in vitro transcribed RNA for NoV GI, NoV GII, HAV, and mengovirus (vMC0). Virus genomic quantification in wastewater samples using qPCR based on the construction of standard curves using plasmid and in vitro transcribed RNA are represented as circles and stars, respectively. The linear regression line was obtained plotting the known quantities of serially diluted standard samples against the cycle quantification (Cq) of the samples. The slopes of the regression line were used to calculate the amplification efficiency (Ef) of the qPCR reactions according to the formula Ef = 10 (−1/slope).

**Figure 3 microorganisms-09-01149-f003:**
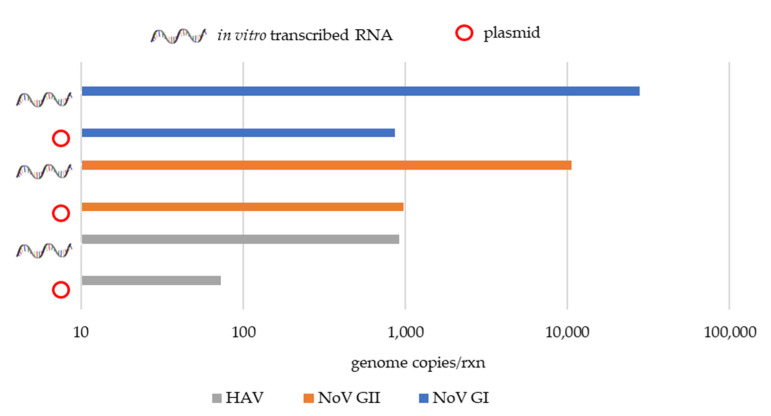
Comparative qPCR quantifications for positive samples to NoV GI, NoV GII and HAV genomes using plasmid and in vitro transcribed RNA standard curves.

**Figure 4 microorganisms-09-01149-f004:**
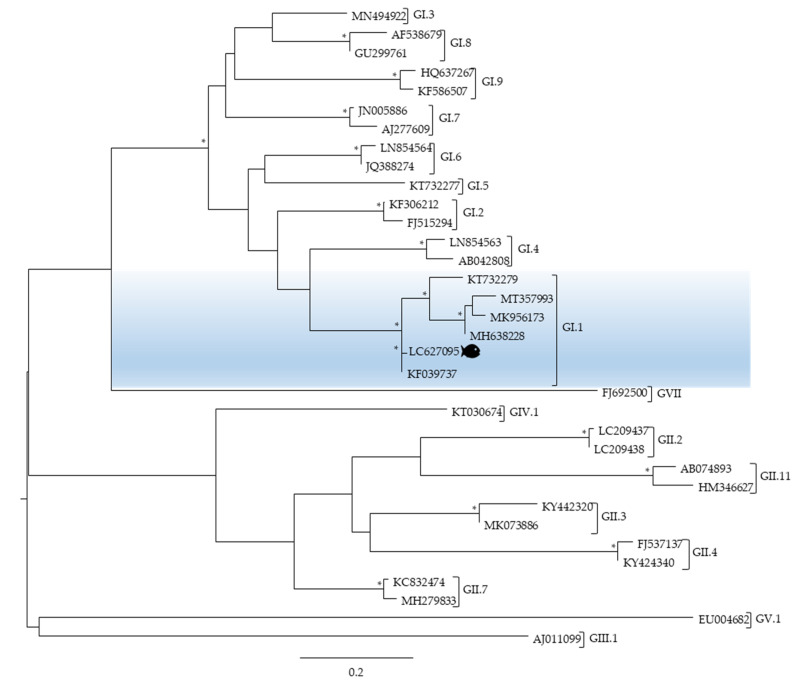
Phylogenetic analysis of the norovirus sequence detected in a pool of brains of seabass caught on the Portuguese coast by targeting part of the gene encoding the open reading frame 1 (ORF1)-ORF2 junction region. All viral sequences used are from human hosts and are identified by their accession number and genogroup; a fish is highlighting the viral sequence identified in this study. Scale bar indicates nucleotide substitutions per site and bootstrap values higher than 75% are displayed by *.

**Table 1 microorganisms-09-01149-t001:** Fish samples used in this study, their source, and fishery type.

Species	Source	Fishery Type	N° of Specimens	N° of Pools
*Trachurus trachurus* (Atlantic horse mackerel)	Figueira da Foz fish market	Wild fisheries	30	3
*Trachurus trachurus* (Atlantic horse mackerel)	Peniche fish market	Wild fisheries	20	2
*Trachurus trachurus* (Atlantic horse mackerel)	Algarve fish market	Wild fisheries	30	3
*Trachurus trachurus* (Atlantic horse mackerel)	Supermarket	Wild fisheries	10	1
*Trachurus trachurus* (Atlantic horse mackerel)	Discarded from fish markets	Wild fisheries	6	1
*Sardina pilchardus* (sardine)	Algarve fish market	Wild fisheries	30	3
*Sardina pilchardus* (sardine)	Sagres fish market	Wild fisheries	30	3
*Sparus aurata* (gilthead seabream)	Algarve fish market	Wild fisheries	15	3
*Sparus aurata* (gilthead seabream)	Algarve fish market	Aquaculture	15	3
*Sparus aurata* (gilthead seabream)	Setúbal fish market	Aquaculture	15	3
*Sparus aurata* (gilthead seabream)	Peniche fish market	Wild fisheries	15	3
*Sparus aurata* (gilthead seabream)	Supermarket	Aquaculture	7	1
*Sparus aurata* (gilthead seabream)	Discarded from fish markets	Wild fisheries	5	1
*Sparus aurata* (gilthead seabream)	Discarded from fish markets	Aquaculture	5	1
*Dicentrarchus labrax* (seabass)	Algarve fish market	Aquaculture	15	3
*Dicentrarchus labrax* (seabass)	Setúbal fish market	Aquaculture	15	3
*Dicentrarchus labrax* (seabass)	Peniche fish market	Wild fisheries	15	3
*Dicentrarchus labrax* (seabass)	Figueira da Foz fish market	Wild fisheries	15	3
*Dicentrarchus labrax* (seabass)	Supermarket	Aquaculture	7	1
*Dicentrarchus labrax* (seabass)	Discarded from fish markets	Aquaculture	4	1
*Merluccius merluccius* (European hake)	Discarded from fish markets	Wild fisheries	6	1
*Mullus surmuletus* (mullet)	Discarded from fish markets	Wild fisheries	5	1
*Mugil cephalus* (rooster)	Discarded from fish markets	Wild fisheries	2	1
*Chelidonichthys lucerna* (redfish)	Discarded from fish markets	Wild fisheries	3	1
*Mugil cephalus* (flathead grey mullet)	Discarded from fish markets	Wild fisheries	3	1

**Table 2 microorganisms-09-01149-t002:** Nucleotide sequences of primers and probes used in this study.

Target	Primers/Probes (5′-3′)	Reference	Reference Sequence
Fw_Mengo (vMC0)	GCGGGTCCTGCCGAAAGT	[16]	L22089
Rv_Mengo (vMC0)	GAAGTAACATATAGACAGACGCACAC		
P_Mengo (vMC0)	ATCACATTACTGGCCGAAGC		
Fw_NoV GI	CCATGTTCCGBTGGATGC ^a^	[17]	M87661
Rv_NoV GI	CCTTAGACGCCATCATCATTTAC	[18]	
P_NoV GI	AGATRGCGATCTCCTGTCCACA ^a^	[18]	
Fw_NoV GII	ATGTTYAGRTGGATGAGATTCTC ^a^	[17]	AF145896
Rv_NoV GII	TCGACGCCATCTTCATTCACA	[18]	
P_NoV GII	TGGGAGGGCGATCGCAATCT	[18]	
Fw_HAV	TCACCGCCGTTTGCCTAG	[13]	M14707
Rv_HAV	GGAGAGCCCTGGAAGAAAG	[13]	
P_HAV	GATTCCTGCAGGTTCAGGGTTCT	This study	
NoV GI_nFw1	CGYTGGATGCGNTTYCATGA ^a^	[18]	M87661
NoV GI_nRv1/2	CCAACCCARCCATTRTACA ^a^	[19]	
NoV GI_nFw2	CTGCCCGAATTYGTAAATGA ^a^	[19]	
NoV GII_nFw1	CARGARBCNATGTTYAGRTGGATGAG ^a^	[18]	AF145896
NoV GII_nRv1/2	CCRCCNGCATRHCCRTTRTACAT ^a^	[19]	X86557
NoV GII_nFw2	CNTGGGAGGGCGATCGCAA ^a^	[19]	X86557
HAV_nFw1	TATGCYGTITCWGGIGCIYTRGAYGG ^a^	[20]	NC_001489
HAV_nRv1	TCYTTCATYTCWGTCCAYTTYTCATCATT ^a^		
HAV1_nFw2	GGATTGGTTTCCATTCARATTGCNAAYTA ^a^		
HAV2_nrv2	CTGCCAGTCAGAACTCCRGCWTCCATYTC ^a^		

^a^ Mixed bases in the primers: B = C/G/T, R = A/G, Y = C/T, N = A/T/C/G, H = A/C/T, W = A/T, I = inosine.

**Table 3 microorganisms-09-01149-t003:** Primers and probes concentrations used in the optimised single and multiplex qPCR reactions.

Reagents	NoV GII, HAV, Mengovirus Single Reaction	NoV GI Single Reaction	HAV, NoV GII Multiplex Reactions	NoV GI Multiplex Reaction	Mengovirus Multiplex Reaction
Reverse primer	900 nM	500 nM	400 nM *	400 nM	900 nM
Forward primer	500 nM	100 nM	100 nM	100 nM	500 nM
Probe	250 nM	250 nM	100 nM	250 nM	250 nM

* in cDNA template the concentration of reverse primer is 500 nM.

**Table 4 microorganisms-09-01149-t004:** Real-time PCR efficiencies and determination coefficients of the optimised standard curves using plasmid and in vitro transcribed RNA as templates.

Plasmid					In Vitro Transcribed RNA			
	Single		Multiplex		Single		Multiplex	
	qPCR Efficiency (%)	R^2^	qPCR Efficiency (%)	R^2^	qPCR Efficiency (%)	R^2^	qPCR Efficiency (%)	R^2^
Nov GI	89.3	0.999	94.4	0.999	90.8	0.999	106.9	0.995
NoV GII	100.8	1	98.0	0.999	97.0	0.999	111.6	0.985
HAV	97.6	0.999	105.5	0.999	88.0	0.997	108.8	0.999
Mengovirus	97.1	0.996	97.6	0.999	97.1	0.996	96.3	0.999

**Table 5 microorganisms-09-01149-t005:** Real-time PCR quantifications for positive samples to NoV GI, NoV GII and HAV genomes using the plasmid and in vitro transcribed RNA standard-based curve.

	Plasmid		In Vitro Transcribed RNA	
Viruses	Cq Mean	Genome Copies/rxn	Cq Mean	Genome Copies/rxn
Nov GI	34.94	8.60 × 10^2^	34.43	2.80 × 10^4^
NoV GII	31.52	9.82 × 10^2^	31.63	1.06 × 10^4^
HAV	33.76	7.30 × 10^1^	33.07	9.18 × 10^2^

**Table 6 microorganisms-09-01149-t006:** Mengovirus genome sequences quantifications and recovery rates in artificially inoculated samples (*n* = 3) using the optimised qPCR assays.

Plasmid				In Vitro Transcribed RNA		
**Tissue**	Amount (in Genome Copies/µL)	Cq Mean	Recovery Rates (%)	Amount (in Genome Copies/µL)	Cq Mean	Recovery Rates (%)
liver	1.12 × 10^6^	29.23	11.18	1.29 × 10^6^	28.92	12.90
	1.58 × 10^5^	32.86	15.78	9.66 × 10^5^	32.91	9.66
	2.95 × 10^4^	35.53	29.46	1.56 × 10^4^	35.91	15.63
gills	2.10 × 10^6^	27.58	21.00	2.74 × 10^6^	27.12	27.37
	1.34 × 10^5^	29.51	13.37	1.42 × 10^5^	29.67	14.21
	3.22 × 10^4^	32.22	32.20	2.24 × 10^4^	32.23	22.40

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
