# Peer review of "Development of a Tetraplex qPCR for the Molecular Identification and Quantification of Human Enteric Viruses, NoV and HAV, in Fish Samples"

_microorganisms, 2021, doi:10.3390/microorganisms9061149_

Round 1

Reviewer 1 Report

In Molecular identification and quantification of human enteric viruses, Nov and HAV, in seabass,  by Filipa-Silva et al the authors establish a qPCR method to detect foodborne infections of human enteric viruses namely norovirus (NoV) and hepatitis A virus (HAV).  The article describes the methodology and validation of the qPCR  assay, its specificity and sensitivity to NoV and HAV.  Using the methodology the authors screen a populations of gilthead seabreams, european seabass two of the most cultured species in the Mediteranean sea as well as sardine and mackerel which are also popular in consumption in Portugal.  The authors aim to assess the the presence of these important foodborne pathogens in the fish assayed.  

The  qPCR assay is properly designed and executed, with both plasmid and cDNA serving as potential standard curves for analysis of genomic copy sensitivity.  The authors establish the specificity of the assay by running against common fish pathogens and introduce and external control for the RNA extraction, RT and amplification in the form of a mengovirus. 

A multiplex assay is established which shows similar sensitivity to the singleplex assay, allowing the use of this diagnostic method in detecting both NoV and HAV in a single assay.  

The results identify a single positive (of 400 pools assayed) brain to NoV.  This is quite a surprising result, as the author state themselves, as no other organs, which NoV has more tropism towards are positive.  Moreover, no other virological methods of isolation and determination of viral presence were carried out to validate this "surprising" finding.   The authors suggest that the detection of NoV in the fish samples is through water contamination, which is plausible.  Yet further in the discussion suggest that the virus itself infected the fish brain, which resulted in the single positive result.  The latter is not supported by the data, and further studies should be carried out to establish whether NoV infects fish.  This would be the first recorded case of a human virus infecting a fish.  

I would consider to remove the conclusion of the virus infecting the fish itself and focus on the established detection method. Also consider the following points: 

  1. Line 80 – “are a relatively” - “are relatively”
  2. Line 91 – “seabreams” – “seabream”
  3. Line 92 – aquacultures – aquaculture
  4. Line 128 – Authors state that “eight tissue samples were selected based on natural tropism”, has it been established that these viruses infect fish ? has the tissue tropism of these viruses in fish been established? These viruses have enteric tropism yet the only positive sample was extracted from a brain of a fish.
  5. Line 210- the use of positive waste water is unclear.
  6. Line 211 – “…genomes were used to validate que qPCRs…” does the “que” mean anything or is this a typo?
  7. Line 216 – Delete “comparative”
  8. Figure 2 – unclear, please elaborate on what it is you are trying to show
  9. When analyzing the cdc website for noro virus related outbreaks during the 2007-2018 period filtered for waterborne origin shows 55 outbreaks, 3330 illnesses and 33 hospitalization cases. One should consider the possibility of waterborne related contamination of the samples, which results in a positive qPCR assay, as only a single positive case was observed. 
  10. The conclusion of a human viral pathogen infecting a fish species is not supported by the results.  Live viral isolation/ In situ hybridization / Electron microscopy of brains positive for Norovirus by qPCR would have given more credence to the data.
  11. One must also consider the possibility of an accidental identification of a fish associated Norovirus which a conserved region has yielded a positive result. Further sequencing of the isolate, beyond the ORF1-ORF2 -junction region would allow a better classification and understanding of the origin of the virus, and whether a novel NoV has been identified.
  12. The authors do not address the tropism of the virus, the only positive result is from a brain of a single sample, yet in other organs where you would expect the virus to propagate all results are negative.
  13. The only positive result, of the 323 specimens assayed are from the Peniche fisheries. Yet in table 1 the source is defined as the Peniche fish market, can this be a result of human handling contamination and not a bona fide infection of the fish itself?

Reviewer 2 Report

General comments

The paper I reviewed “Molecular identification and quantification of human enteric viruses, NoV and HAV, in seabass” represent a study aimed to develop a tetraplex qPCR assay for the detection and quantification of Norovirus (NoV) GI, NoV GII, and HAV genomes using plasmid or an in vitro transcribed RNA for standard curve construction, comparing the two standards curve types in terms of efficiency, sensitivity, and detection limit. The Authors observed for the multiplex qPCR with the in vitro transcribed RNA-based standard curve a higher sensitivity in approximately 1 log unit compared with the qPCR using the plasmid-based standard curve. By using tetraplex qPCR assay to perform a virological screening of 323 fish specimens captured and farmed along the Portuguese coast, a NoV GI genome was identified in a pool of seabass brain tissue, representing the first report.

.

The article is well written and interesting. The statements described are supported by detailed presented data. I have only some observations.

1)I suggest to the Authors to modify the title since it doesn’t express the real impact of the work for the development of the qPCR, also comparing the two standard curve types. Furthermore, reading this title I think that NoV and HAV were detected in seabass, but it is not the result obtained.

2)Line 194-207: it is not clear the subdivision of the two paragraphs 2.6 and 2.6.1 with the same title.

3)Line 316: remove the repetitive “Comparative”.

4)I suggest to the Authors to explain, at least in the figure, the fragment used for the phylogenetic analysis.
